# Ganglionated Plexus Ablation Procedures to Treat Vasovagal Syncope

**DOI:** 10.3390/ijms241713264

**Published:** 2023-08-26

**Authors:** Merav Yarkoni, Wajeeh ur Rehman, Ata Bajwa, Alon Yarkoni, Afzal ur Rehman

**Affiliations:** Heart and Vascular Institute, United Health Services, Johnson City, NY 13790, USA; wajeeh.rehman9@gmail.com (W.u.R.); atabajwa@yahoo.com (A.B.); alonyarkoni@gmail.com (A.Y.); afzal.rehman@nyuhs.org (A.u.R.)

**Keywords:** vasovagal syncope, cardio-neuroablation, GP, ganglionated plexus, ablation, vagal nerve, spectral analysis, anatomical approach, high-frequency stimulation, HFS, cardio-neuromodulation

## Abstract

Vasovagal syncope (VVS) refers to a heterogeneous group of conditions whereby the cardiovascular reflexes normally controlling the circulation are interrupted irregularly in response to a trigger, resulting in vasodilation, bradycardia, or both. VVS affects one-third of the population at least once in their lifetime or by the age of 60, reduces the quality of life, and may cause disability affecting certain routines. It poses a considerable economic burden on society, and, despite its prevalence, there is currently no proven pharmacological treatment for preventing VVS. The novel procedure of ganglionated plexus (GP) ablation has emerged rapidly in the past two decades, and has been proven successful in treating syncope. Several parameters influence the success rate of GP ablation, including specific ablation sites, localization and surgical techniques, method of access, and the integration of other interventions. This review aims to provide an overview of the existing literature on the physiological aspects and clinical effectiveness of GP ablation in the treatment of VVS. Specifically, we explore the association between GPs and VVS and examine the impact of GP ablation procedures as reported in human clinical trials. Our objective is to shed light on the therapeutic significance of GP ablation in eliminating VVS and restoring normal sinus rhythm, particularly among young adults affected by this condition.

## 1. Introduction

Vasovagal syncope (VVS) is the most common form of neurally mediated reflex syncope (NMS), and usually refers to a heterogeneous group of conditions whereby the cardiovascular reflexes normally controlling the circulation are interrupted irregularly in response to a trigger. This results in vasodilation, bradycardia, or both, thereby promoting syncope [1]. Also known as reflex neurocardiogenic syncope, VVS is the most frequent cause of a temporary loss of consciousness, especially in young adults without a substantial heart disease [2]. VVS is a frequent clinical problem affecting one-third of the population at least once in their lifetime [3], or by the age of 60 [4,5]; it reduces the quality of life, and may cause disability affecting certain routines. It poses a considerable economic burden on society [6], and, despite its prevalence, there is currently no proven pharmacological treatment to prevent VVS [7]. The placement of dual-chamber pacing is effective in only a subgroup of VVS patients with significant asystolic NMS [8]. The novel procedure of ganglionated plexus (GP) ablation has developed rapidly in the past two decades, and has been proven successful in treating syncope. Several parameters influence the success rate of GP ablation, including specific ablation sites, localization and surgical techniques, method of access, and integration with other interventions. This review aims to provide an overview of the existing literature on the physiological aspects and clinical effectiveness of GP ablation in the treatment of VVS. Specifically, we explore the association between GPs and VVS and examine the impact of GP ablation procedures as reported in human clinical trials. Our objective is to shed light on the therapeutic significance of GP ablation in eliminating VVS and restoring normal sinus rhythm, particularly among young adults affected by this condition.

## 2. Cardiac Innervation

The heart is innervated by nerves from the brain and spinal cord, which either stimulate or inhibit physiological cardiac functions such as heart rate and contraction force. These nerves make up the cardiac autonomic nervous system (CANS) and include sympathetic and parasympathetic avenues. Under pathological conditions, the density of these nerve fibers in the heart may be altered, leading to either too much activation (hyper-innervation) or too little (hypo-innervation). This difference in innervation may cause a spatial imbalance in the activation of the heart, leading to cardiac autonomic dysfunction [9] related to the development and progression of cardiovascular diseases such as myocardial infarction (MI), arrhythmias, hypertension, heart failure [10,11,12,13], and several types of syncope [1,3].

Cardiac contraction and heart rate are achieved by the innervation of heart tissue via CANS. All neuronal elements, including cell bodies, axons, and dendrites, are arranged and bundled to form nerves; the sympathetic and parasympathetic nerves that comprise CANS originate in the central nervous system (CNS) as preganglionic nerves, while sensory afferent nerves originate in the heart tissue and blood vessels and project into the CNS. The difference between the sensory and autonomic nervous systems is the location of their ganglia, where synapses of the postganglionic neurons derive. Innervation of the heart is established via postganglionic synapses, where acetylcholine neurotransmitters bind to postsynaptic receptors located on the postganglionic neuron inside the heart.

Sensory nerves have a different anatomy and function than the sympathetic and parasympathetic nerves. Their main cardiac function is to provide feedback from baroreceptors on the aortic arch to the brain to maintain homeostasis. Sensory nerves provide information to the CNS and convey information on blood pressure, oxygen, carbon dioxide, and sugar levels in the blood. The sensory cell body gives rise to axons which sprout to both the heart and brain. The signal starts in the heart and is directed towards the brain via the neuronal cell body, whereas in sympathetic and parasympathetic nerves, the cell body also indicates the starting site of the signal. The sensory cell body lies within the dorsal root ganglia in the peripheral nervous system (PNS). Sensory nerves lay in close proximity to the autonomic nerves, and while responding to the cardiac stimulus, they relay back to the brain. Subsequently, the brain modulates output towards the heart, which is regulated by the parasympathetic and sympathetic nerves.

The cardiac autonomic nervous system can be simply divided into extrinsic and intrinsic elements according to the alignment and localization of nerve fibers, their cell bodies, and joint ganglia. The extrinsic, or central, CANS comprises central nerves originating in the brain and the spinal cord and forming the external ganglia, which then synapse with secondary neurons en route to the heart. The intrinsic CANS is confined within the heart and specifies small parasympathetic neurons that synapse with vagal sprouting nerves to activate their responses within the heart. The intrinsic system designates an extensive epicardial neural network comprising clustered nerve fibers of all types and parasympathetic cell bodies, which together constitute the GP, on both the atria and ventricles, as well as around pulmonary veins and great vessels [14]. These GPs, usually embedded within epicardial fat pads, vary in size from very few neurons to containing over 400 neurons [15,16]. Notably, the most dense autonomic innervation is localized at the posterior wall of the left atrium, predominantly at the pulmonary vein–atrial junction [16].

## 3. GP Anatomy and Physiology

The GPs are composed of a heterogeneous population of neurons, including efferent (into the heart), afferent (away from the heart), and many small interconnecting neurons comprising the bulk of the GP [17]. Additionally, GPs encompass both sympathetic and parasympathetic neural elements, and function as communication centers between the intrinsic and the extrinsic CANS, managing and controlling the electrophysiological, vascular, and contractile functions [18].

### 3.1. Anatomic Location of GPs

Anatomically, the four major atrial GPs are located in close proximity to the pulmonary veins (PVs), and each innervates one of the four PVs, as well as the surrounding atrial myocardium [15,16] (Figure 1). In addition, the fifth left lateral GP (LLGP) is located around the ligament of Marshall. These GPs can be identified during electrophysiological studies by applying high-frequency stimulation (HFS) (20 Hz) at the respective anatomical locations [19,20,21,22,23,24]. A positive response is defined as an increase in the R–R interval by >50% during atrial fibrillation (AF) [19]. The GPs were thus named based on their anatomical locations [19]. The complex anatomical layout and physiological interconnectivity of these GP sites is important to understanding the pathophysiology of VVS. Knowledge of GPs’ locations and their axonal projection pathways is important when considering targeted therapeutic interventions.

### 3.2. Physiology of GPs

As previously described, the CANS consists of parasympathetic, sympathetic, and sensory neural elements residing in different anatomical sites of the heart. It is recognized that parasympathetic and sympathetic systems reach their target organs via pre- and postganglionic nerve fibers. The cell bodies of postganglionic sympathetic and sensory neurons lie in the CNS and away from the heart, whereas cell bodies of postsynaptic parasympathetic neurons are very close to or within the heart (Figure 2). This distinct neuroanatomy facilitates selective vagal denervation, because only local postganglionic neuronal cell bodies are abolished by endocardial radiofrequency energy [25,26]. This is termed cardiac ablation, causing irreversible damage to vagal efferents, while postganglionic nerve fibers of sympathetic and sensory neurons are preserved, as they may be repaired by axonal regeneration [27,28] (Figure 2).

### 3.3. The Vagal Nerve

Under normal physiological conditions, the involuntary function of the heart is maintained by the balance of inhibitory (parasympathetic, vagal nerve) and excitatory (sympathetic) tones. This balance is facilitated by the brainstem and regulates the instantaneous heart rate. In certain instances, this balance is impaired due to an over-stimulated vagal response triggered by the vagal reflex. Physiologically, the vagal reflex, which can activate through multiple triggers, produces (1) severe cardioinhibition (causing asystole, serious bradycardia, and/or temporary complete AV block) and/or (2) vasodilation (the vasodepressor form, leading to critical and transient hypotension). In the case of cardioinhibition, which instigates prolonged asystole, the patient typically recovers spontaneously within minutes.

## 4. Neurophysiology of Ablation Targets

The mechanism of vagal stimulation is initiated by the local release of acetylcholine. There is an influx of acetylcholine stimulating the PV ganglia, inducing PV firing and reducing the action potential duration in the atrial myocytes in the PV sleeves, causing them to fire until suppressed [29], and thus maintaining AF. Studies have suggested that the serotonin signaling pathway is also over-regulated in VVS [30].

Several animal studies have shown that vagal efferent postganglionic neurons innervating the heart are primarily located in discrete epicardial ganglionic structures known as fat pads [31,32,33]. Subsequently, at these sites, emerging human studies have aimed to permanently denervate the vagal response by means of radiofrequency catheter ablation. The first studies were related to vagal-induced AF [34,35]. To define the exact localization of these fat pads, two different approaches were used: HFS-induced vagal reflex followed by atrial cardiac ablation to prevent AF [34] and fast Fourier transform (FFT) analysis to define GPs [35]. Good GP ablation resulted from a canine model of AF [36], promoting practitioners to target GP with ablation in patients.

Pachon et al. [37] were the first to propose specific vagal denervation by catheter ablation and spectral mapping for different arrhythmias and severe cases of cardioinhibitory syncope, giving rise to cardio-neuroablation (CNA). Since then, practitioners have used CNA to completely abolish or significantly reduce the vagal response, eradicating symptoms in >75% of patients, with limited complications for 14 years. As a result, CNA became an effective substitute to treat syncope syndromes in young patients without requiring a pacemaker implantation. The first patient was ablated in 2002 via CNA of the vagal nerve as the ultimate remedy for severe VVS [2]. Since then, significant improvements followed, and many remarkable authors have been able to reproduce the results [19,38,39,40,41,42,43]. Accordingly, in the last 20 years, catheter cardiac ablation has become a recognized, essential treatment strategy for VVS.

## 5. GP Ablation as the Fundamental Treatment for VVS

Several strategies have emerged to treat VVS by CNA and are currently used to target GP inputs within the heart. This evolving comprehensive approach includes anatomic and electrocardiographic mapping techniques, as well as physiologic and pharmacologic methods. The physiologically-guided approaches incorporate spectral mapping or HFS, whereas anatomically-guided methodologies purely target the GPs by location. Currently, there is no consensus on the best tactic for identifying atrial GP sites for CNA. Several additional parameters have not yet been confirmed; these include (1) the extent to which GP ablation is necessary to attenuate the vagal activity in patients, and (2) corroboration of the vagal denervation endpoints [44].

Three subtypes of VVS can be divided into the following broad categories: (1) pure cardioinhibition (with or without asystole); (2) pure vasodepression, in which hypotension occurs without a significant decrease in heart rate; and (3) a mixed type of cardioinhibition and vasodepression. Cardioinhibition is caused by abrupt and intense vagal reflex with or without defined triggers. Despite pharmacological and pacemaker implantation efforts, the latter are highly rejected by young patients. The Second Vasovagal Pacemaker Study confirmed the weak evidence of the efficacy of pacemaker therapy [45].

## 6. Ganglia Detection Methods

Several different approaches have been used for identification of parasympathetic autonomic ganglia in atria, including HFS, spectral analysis (SA), and an anatomical approach (AA). We comprehensively examine them herein and discuss the pros and cons of each. Prior to the use of ablation as the principal method to denervate the CANS, the achievement of pulmonary vein isolation (PVI) served as a suitable procedure. PVI is associated with denervation of the CANS and a significant reduction in AF recurrence [46]. Endocardial and epicardial access during PVI procedures has been associated with unintentional damage and incidental ablation at GP sites [47]. PVI, via thermal epicardial approaches, can result in the overlap of ablation lesions with numerous GP sites, while endocardial thermal approaches may induce collateral damage by conductive heating. One downfall to this technique is that conventional PV antrum isolation often unknowingly involves ablation of autonomic nerves positioned at the right anterior GP (RAGP), left superior GP (LSGP), and ligament of Marshall; however, the left inferior GP (LIGP) and right inferior GP (RIGP) remain largely intact [19]. Thus, Katritis et al. showed that >50% of their paroximal AF patients had recurrent syncope after 12 months of follow-up [48].

As opposed to the PVI technique [48,49], GP ablation can be performed either empirically, at their presumed anatomical locations [21,50,51], or the GP can be identified by applying HFS, as described by the Oklahoma group [19] and others [19,20,21,22,23,24] (Table 1 and Table 2). The Oklahoma group suggested that the GPs function as “integration centers” between extrinsic and intrinsic cardiac CANS [19].

### 6.1. Spectral Mapping Analysis (SA)

In 2004, Pachon et al. [35] used the SA of the atrial potentials, by which this group of researchers recorded and studied the whole frequency spectrum of the endocardial potentials by the FFT. Using this methodology, they found two kinds of atrial myocardium, the compact and the fibrillar [35,37]. The compact myocardium displayed a homogeneous spectrum with one main frequency around 40 Hz and uniform conduction which resulted from a mass of very well-connected cells [35,76,77]. The fibrillary myocardium appeared as a heterogeneous, rough, and fractioned spectrum with frequencies higher than 100 Hz, shifting its FFT to the right [35,76,77]. This conformation of cell bundles working in grouped filaments, as nerve fibers interweave with myocardial cells, alters myocardial conduction from compact to the fibrillary pattern [35,76,77]. The concurrent mixture of cells changes some electrical properties of the atrial wall and can easily be detected by deviations in the spectrum as “AF Nests” [40,41,42,43]. Thus, the distinguished endocardial fibrillary pattern was used as a marker of the neuro-myocardial interface. The fibrillary myocardium predominantly encompasses the anatomical regions of the cardiac GP [35,77,78], and, therefore, characterizes these sites as AF nests.

By applying online, real-time spectral mapping, it is possible to reveal the fibrillary myocardium (AF nests) to lead the ablation of the first neuron [79]. Using this method, nearly all postganglionic parasympathetic neurons may be eliminated and may not recover, whereas the sympathetic and sensory terminal fibers usually recuperate within weeks to months [79].

In 2011, Pachon et al. performed extensive right and left atrial ablation to treat patients with VVS, guided by spectral analysis for “AF nests” to complete vagal denervation. In their study, 40 of 43 patients remained free of syncope after a mean follow-up of 45 months [52]. Although never used as a stand-alone strategy to guide ablation, this technique has been attempted in combination with empirical anatomic ablation, HFS, or both by distinct research groups [37,38,39,52,55,70,71].

### 6.2. Anatomically-Guided Approach (AA) [21]

As a stand-alone tactic, the AA was initially performed in two VVS cases by Rebecchi et al. [69]. In this study, the authors achieved practical ablation through the right atria and around three major ganglia up until all atrial electrical activity was fully exterminated. A similar approach was used by Debruyne et al. [72] in a 16-year old female suffering from VVS. In their work, only the superior right atrial ganglion was targeted and ablated using a multi-electrode irrigated ablation catheter. A computed tomographic scan characterizing the cardio-neuromodulation (CardNM) methodology (discussed below) was also used in this study.

In the same year, Sun et al. [21] compared 10 individuals who underwent HFS-guided ablations (see below) vs. 47 individuals undergoing AA ablations, all of whom had VVS (Table 1). In the HFS-guided group, ablation was performed by selective vagal denervation, as previously described in Yao’s study [23] (see below). In the AA group, the four previously defined parasympathetic autonomic ganglia (PAG) were targeted, and the radiofrequency energy was initially delivered to induce a vagal response (VR) within 10 s; further energy was delivered for at least 30 s until complete inhibition of VR was achieved. Further ablation performances adjacent to the initial lesion were implemented to produce a cloudlike lesion pattern until five repeated ablation efforts failed to induce additional VR.

In the following year, in 2017, AA was used by Rivarola et al. [50] in a heterogeneous group of patients suffering from VVS, advanced AVB, or sinus arrest. They targeted six experimentally defined sites based on the combination of ganglion localization methods used in Yao’s (HFS) and Rebecchi’s (AA) studies [23,69]. In that year, Qin et al. prospectively studied AA ablations in 62 patients < 50 years old with symptomatic sinus bradycardia (SB). They were able to show that all patients presented with significantly improved symptom outcomes one year after ablation. Very recently, in 2021, two studies were conducted on VVS patients using the AA methodology [51,60].

### 6.3. High Frequency Stimulation (HFS) Approach

Po et al. developed a technique to localize and ablate the GP, which serves as the “integration centers” of the intrinsic cardiac autonomic nervous system (ICANS) [19]. Applying HFS to the atria causes two modes of response: (1) VR, defined as a major prolongation of the P-R or R-R intervals; or (2) a normal response, characterized by no or insignificant fluctuations on the PR or RR intervals. These two reactions reveal vagal innervation sites and normal atrial myocardium, respectively. This group of researchers identified the four major atrial GP sites by delivering HFS (20 Hz, 10–150 V, 1–10 ms pulse width) to atrial tissue where GPs were postulated to be localized. Locations displaying a parasympathetic response, which they randomly classified as a ≥50% increase in the mean RR interval during AF, was allocated as a GP site. A subsequent radiofrequency current was applied at that site to abolish the parasympathetic response [19].

The first group to practice selective HFS-guided GP ablation for the treatment of VVS was introduced by Yao et al., who reported successful outcomes in 10 patients with highly symptomatic VVS [23]. Selective HFS-guided GP ablation resulted in an impressive amelioration of their prodromal symptoms and no recurrence of syncope over a 30-month follow-up period [23]. The same group recently reported their long-term results of GP ablation in a larger cohort of patients with VVS [21], as 57 consecutive patients with highly symptomatic VVS received either anatomical sequential ablation of all four major atrial GPs in their recognized anatomical locations or HFS-guided GP ablation. During a mean 36-month follow-up, 91% of patients remained syncope-free, and prodromes were significantly reduced [21]. There were no differences between anatomical and HFS-guided GP ablation in terms of syncope or prodromes [21]. The HFS technique to guide ablation of the PAG for VVS showed benefits in three cohort studies [21,23,53] and four case reports [20,40,41,66].

### 6.4. Cardio-Neuromodulation (CardNM) Approach

CardNM is a novel approach which was instigated in recent years by Debruyne et al. [72] in order to achieve a faster procedure, minimizing procedural risks in patients compared to the conventional ablation strategies of left-sided or bi-atrial multiple sites [65]. The group demonstrated that it is possible to treat NMS, functional sinus bradycardia, and vagal-mediated AVB by performing CardNM [64,65,72,80], but not functional AVB [65]. The principal strategy of this technique is to approach the left endocardium by targeting the ARGP. To avoid exposing patients to additional unnecessary risks, the investigators believed that CardNM via the right-sided approach should be the first step for all NMS patients referred for ablation, and that targeting the IRGP methodologically should not be performed [81]. To avoid entering the left atrium, the authors used computed tomography and fusion imaging to restrict the procedure to the right heart cavities [80]. Indeed, the mean ablation time for these procedures is significantly shorter than most other CNA performances [64]. The long-term clinical results of CardNM are good, with persisting adequate partial sinus node vagolysis [65].

Qin et al. further confirmed that targeting the ARGP and the aorta-superior vena cava (Ao-SVC) fat pad (using the AA approach) is sufficient to terminate the vasovagal response, as the sinus rate (SR) increased considerably at these particular sites promptly post-ablation [61]. However, ablation to other areas, namely, SLPG, ILGP, and IRGP, did not substantially increase the mean SR in young patients (<50 years old) suffering from symptomatic SB [61]. This fact alone can potentially decrease the ablation time drastically, and should also minimize cardiac bruising.

### 6.5. Fractionation High-Density Mapping

The new methodology of high-density mapping in combination with an auto-algorithm fraction mapping system, developed by Aksu et al. to treat AF and VVS [54], may allow for better and faster localization of GPs during ablation, as a complement to anatomically-guided approaches. Recent studies investigating the treatment of VVS using this method [73,74] have shown improved results in these patients [54] compared to combined spectral analysis with HFS-mapping approaches (Table 1 and Table 2). Additional studies using this mapping strategy, in addition to longer follow-ups for possible VVS recurrence, are still lacking.

## 7. Routes to Access Ablation

As mentioned above, there are several anatomical sites at which to approach the ablation, but the main ones are through the left or right atrium, or both, and sometimes also through the superior vena cava (SVC). Very seldom, other tactics are used. Authors have discussed the pros and cons to each approachable anatomical site, and we summarize their justifications here.

The right atrium GP controls the function of the SA node; electrical stimulation of the RAGP reduces the SR [82]. The SLGP of the left vagosympathetic trunk traverses with the RAGP before proceeding to the SA node; however, Qin et al. showed that RAGP may serve as a “gateway” through which the cardiac autonomic nervous system can modulate the sinus node [61]. Furthermore, the authors showed that ablation of the Ao-SVC GP, which integrates most of the efferent vagal fibers to the atria and then projects onto the RAGP and RIGP [33], suppresses the effects of vagal stimulation on SR slowing [60] and increases the SR [24,61]. Thus, the authors suggest that there is a RAGP-independent and a RAGP-dependent pathway [61].

## 8. Post-Ablation Validations

Several useful methods have been applied in order to evaluate the completion and success of the ablation. Extracardiac vagal stimulation remains the best tool with which to identify and achieve the procedural endpoint [83], and may more accurately distinguish patients who are truly vagally suppressed following CNA [66]. The atrophine test immediately following the ablation, as well as the tilt-test in the chronic phase post-neuroablation, are used to assess the amount of reinnervation and the reappearance of the VR [52]. Heart rate variability (HRV) is another measure used long-term through the 24 h Holter recording. HRV is affected by age, sex, physical fitness, clinical conditions, rest alertness, sleep, and medication [84,85,86], which are all stable parameters during the 24 h testing period.

As the gold standard, the 24 h SDNN (standard deviation of all normal R-R intervals) predicts morbidity and mortality in high-risk and comorbid patients, where SDNN values < 50 ms are classified as at risk, 50–100 ms as compromised health, and >100 ms as healthy [87]. For CNA procedure success, practitioners use the 24 h Holter for SDNN detection, which markedly decreases due to reduced vagal response after the ablation. This apparent reduction, triggered by denervation of the parasympathetic tone, should be distinguished from the declining SDNN values that predict cardiac risk [87]. Herein, we indicate the success of each reported ablation by HRV *p*-values pre- and post-ablation (Table 1 and Table 2) during the period of follow-up.

## 9. Discussion

GP ablation appears to be an efficacious technique for improving outcomes, primarily in patients with neurocardiogenic syncopes. Nonetheless, some very important questions remain unanswered. The long-term outcomes of GP ablation, the precise location and depth of GPs, and the ultimate mechanism of GP ablation to improve VVS are still not fully understood. A recent study by Xu et al. validated that HFS and AA are both efficient and safe approaches to successful VVS relief [57].

Collateral damage is a major disadvantage of current thermal ablation techniques, and introduces a risk of damaging the myocardium and the surrounding tissue. Alternative, less damaging methods are currently available to treat paroxysmal and persistent AF, but do not fix the physiological effects produced by GPs during VVS. The pulsed-field ablation technique is tissue-selective, only damaging myocardial cells [88] and leaving vagal nerves intact and activated. This technique uses non-thermal high-voltage energy to kill cardiomyocytes [89], but not the parasympathetic and sympathetic nerves.

Another problem is the difficulty of delivering precise, appropriate energy to GPs. While GPs, in association with PVs, are accessed with relative ease, others are found in concealed locations. Overall, there is significant complexity involved in catheter positioning both within the pericardial space and the heart itself. A more efficient and effective visualization of GPs using imaging techniques may provide useful information for improved localization. These advances have significantly propelled research over the last decade. While the understanding of GP locations is sometimes obscure, this may be due to the degree of anatomical variability between individuals [47].

However, the specifics in terms of report accuracy of the GPs that have been targeted in some research papers remain ambiguous, with some studies not including nor clearly describing which GPs were targeted or where they were ablated [19,51,62,63]. This causes difficulty when comparing results from different studies targeting specific GPs associated with the maintenance of neural pathways and their subsequent effects on the SA and AV nodes [22,90]. Due to the different techniques and study designs in the literature, it is difficult to assess and make a true comparison regarding success. A recent study confirmed that the order of the targeted GPs affected the occurrence rate of VR and BP reduction during cardio-neuroablation [58]. The RAGP was a critical target for increasing HR and inhibiting VR during the procedure; thus, it may be an integration center in the regulation of other GPs of the LA [58,59].

HRV has been found to be a predictor of ablation success, and is a useful, non-invasive tool for investigating cardiac autonomic tone [91,92]; it measures the fluctuations of time intervals between consecutive heartbeats [85]. It is evident that sufficient disruption of VR results in increased HR and AF relief. It is still possible that post-procedural efficacy may be short-term, due to incomplete ablation of the GPs and the regeneration and formation of new re-entrant pathways in the proximity of the GP.

Recurrent syncope episodes post-CNA are not well described in the literature. Long-term follow-up studies are needed to determine whether nerve regeneration leads to the recurrence of syncope symptoms. A recent report suggested that reinnervation of GPs, in some cases, may be the cause of recurrent syncopes 7 months post-ablation [66]; further indications are warranted. Needless to say, the quality of life post-ablation in patients with previous VVS was improved significantly after a 1-year follow up [93].

Currently, sample size is a major limitation in many studies, with numbers ranging from individual case studies to research including up to 123 patients [63]. Evidently, variation in patient population significantly influence percent outcomes, making it difficult to draw accurate comparisons. From the expansive research and meta-analyses performed to study GP ablation, most results show relief from VVS, both initially and in the long term [21,50,51,55,60,62,63]. While much work is still needed in order to achieve consistency between experiments, it is evident that the potential exists for significant advances in the treatment of VVS through targeted ablation of GP sites.

The use of GP ablation as an adjunctive technique to enhance the outcomes of PVI isolation in patients with paroxysmal AF appears to be both safe and effective. Potential complications associated with GP ablation may rarely occur, especially after LA procedures, such as cardiac perforation, sinus node dysfunction, AV block, phrenic nerve paralysis, and stroke. Two patients (out of sixty-six) developed postoperative bradycardia following HFS-directed CAN, but quickly recovered with drug treatment [57]. In another case of AF, where the AA-CAN was approached from the RA, symptomatic inappropriate sinus tachycardia had occurred in the patient, but completely resolved after 1 month of therapy with a low dose of β-blockers [94]. Finally, two patients experienced sinus node artery occlusion after RA followed by LA-CAN [95].

Encouraging results have been observed regarding GP ablation in patients with VVS. However, several important questions remain unanswered. Firstly, the long-term outcomes of GP ablation, particularly over a period of 5 years, have not been thoroughly studied. Secondly, the precise mechanism through which GP ablation leads to improved outcomes is not fully understood. To address these uncertainties, it is recommended that a randomized, preferably blinded, placebo-controlled trial be conducted to elucidate the true role of GP ablation in VVS patients. Furthermore, the correlation between the clinical outcomes and the extent or technique of ablation remains unknown. Lastly, CNA is an innovative and promising therapeutic option for certain patients with VVS. Existing observational evidence highlights significant success in eliminating syncope in this population, and the occurrence of recurrent syncope following CNA has not been extensively described.

## 10. Conclusions

Several strategies for atrial autonomic denervation have been reported. However, the optimal technique for identifying atrial GP sites, either physiologically guided (SA, HFS) or anatomically guided, has not been established yet [23,37,41,50,52,82]. Further, there is no consensus on the extension of GP ablation for attenuating vagal activity for the treatment of functional bradycardias and VVS, whether ablating all atrial GPs, selectively ablating GPs on both sides of the inter-atrial septum, or exclusively targeting specific areas of the RA or LA GP [21,23,37,39,41,42,50,52,71]. In contrast to pharmacological therapy and pacemaker implantation, GP ablation is designed to target the fundamental problems, which are disturbances in the ICANS. Although the details of CNA required to treat VVS remain to be determined, the results presented in this review indicate, collectively, that GP ablation is an effective and safe treatment option for patients with refractory VVS. This novel technique should be evaluated in large-scale, randomized, controlled trials to demonstrate the efficiency, reliability, and appropriate patient selection criteria.

## Figures and Tables

**Figure 1 ijms-24-13264-f001:**
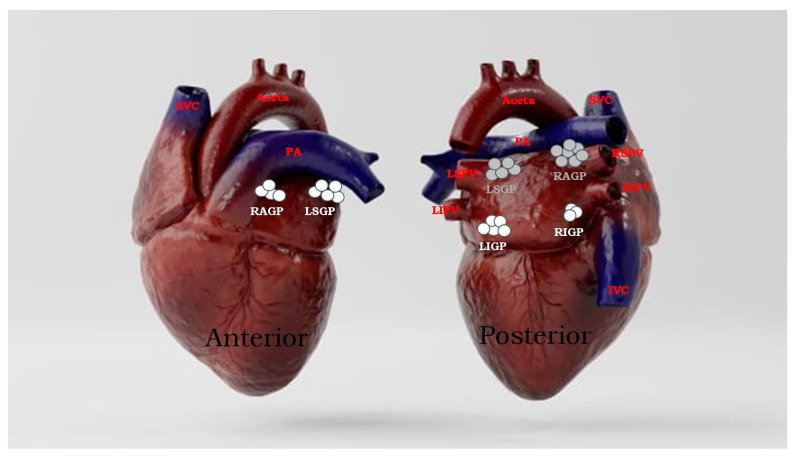
A 3-dimentional representation of the four anatomical GP locations. White (anterior) and grey (posterior) circles indicate GP areas within the heart. (1) The right anterior GP (RAGP) is located on the posterior superior surface of the RA, adjacent to the junction of the superior vena cava (SVC) and RA, close to the right superior pulmonary vein (RSPV). (2) The left superior GP (LSGP) is found on the posterior surface of the LA between the left pulmonary veins (PVs). (3) The right inferior GP (RIGP) is situated on the posterior surface of the RA, adjacent to the interatrial groove. (4) The left inferior GP (LIGP) is located on the posterior medial surface of the LA, close to the left inferior pulmonary vein (LIPV). The left lateral GP (LLGP, not shown) is situated in the area around the ligament of Marshall and can be viewed with a left lateral view.

**Figure 2 ijms-24-13264-f002:**
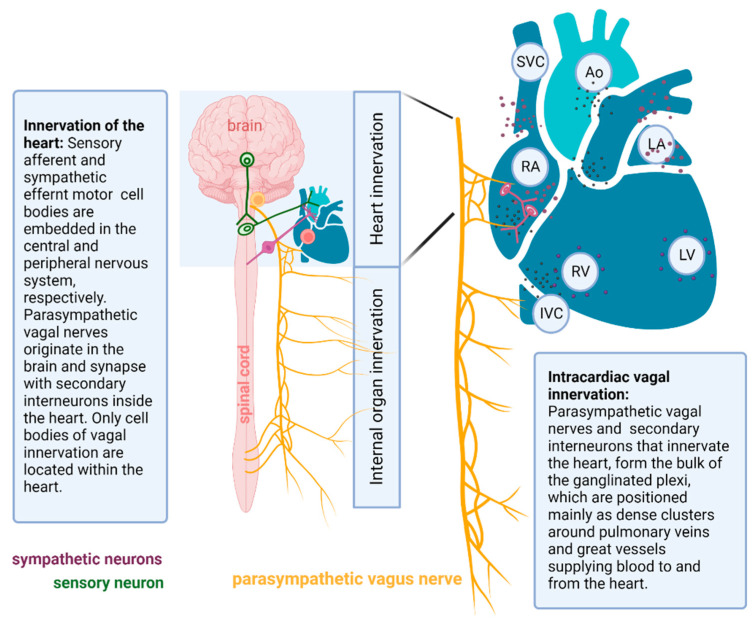
Innervation of the heart includes the afferent (sensory) neurons (green), which transmit information from baroreceptors in the heart to the spinal cord and brain (CNS), and efferent (motor) neurons, which innervate the heart. These efferent motor neurons comprise the cardiac autonomic nervous system, and includes the sympathetic (purple) and parasympathetic (yellow) neurons. Depending on the location of neuronal cell bodies of each neuron type, cardiac innervation is divided into intrinsic and extrinsic elements. Intracardiac vagal innervation (pink) originates in the brain and relays information into the heart, whereas vagal neurons (yellow) synapse with secondary interneurons (pink) to activate the parasympathetic tone. The cell bodies of interconnecting neurons and neural elements of other cell types all cluster into what are known as the ganglionated plexi (small dark dots) in specific cardiac locations with diverse densities. Ao, aorta; IVC, inferior vena cava; LA, left atrium; LV, left ventricle; RA, right atrium; RV, right ventricle; SVC, superior vena cava.

**Table 1 ijms-24-13264-t001:** Previous cohort studies of catheter ablation for neurally mediated syncope.

Technique	*N*	Age (y)	Diagnosis	Approach	Procedure Time (min)	Follow-Up Duration (months)	Syncope Recurrence	Heart Rate Variability (*p*-Value)	Year, Reference
SA + AA *	6	47.5 ± 16	NMS	LA, RA, SVC	38.9 ± 15.4	9.2	0		2005 [37]
SA + AA *	7	47.5 ± 16	Functional high degree AVB	LA, RA, SVC	38.9 ± 15.4	9.2	0		2005 [37]
SA (FFT) + AA *	13	47.5 ± 16	Sinus node dysfunction	LA, RA, SVC	38.9 ± 15.4	9.2	0	0.003	2005 [37]
SA + AA	43	32.9 ± 15	NMS	SVC, RA, inferior–posterior interatrial septum	NR	21.7 ± 11	3 (6.9%)	0.001	2011 [52]
SA (FFT) + HFS *	7	42.7 ± 14.7	AVB	RA in 6 patients, followed by LA in 1 patient	121.2 ± 16.4 (for all patients in this study)	6	1 (14.3%)	NR	2016 [53]
SA (FFT) + HFS *	8	42.7 ± 14.7	NMS	LA then RA	121.2 ± 16.4 (for all patients in this study)	12.3 ± 3.4	0	0.001 (SDNN 6 months post procedure)	2016 [53]
SA (FFT) + HFS *	7	42.7 ± 14.7	SND	LA then RA	121.2 ± 16.4 (for all patients in this study)	9.5 ± 3.1	0	0.001 (SDNN 6 months post procedure)	2016 [53]
Fractionation * Mapping	12	NR	VVS	NR	NR	12	0	SD	2019 [54]
SA + HFS *	8	NR	VVS	NR	NR	12	2	SD	2019 [54]
SA + AA	83	47.3 ± 17	AF VVS	RA, LA	237.2 ± 38	40	0	0.001	2020 [55]
PVI	67	53.2 ± 11.3; 55.2 ± 11.6	Paroximal AF	LA	105.2 ± 22.3; 126.0 ± 26.7	12	18 (54.5%)	NR	2011 [48]
HFS + PVI	83	NR	AF	NR	NR	22	NR	NR	2009 [19]
HFS (selective vagal denervation)	10	50.4 ± 6.4	Recurrent NMS	LA	50.2 ± 3.8	30 ± 16	0	0.002	2012 [23]
Synchronized HFS *	20	50–69	Paroximal AF, AVB	LA	NR	NR	NR	NR	2013 [22]
Continuous HFS *	10	54–68	Persistent AF, AVB	LA	NR	NR	NR	NR	2013 [22]
HFS	11	45.9 ± 10.9	Symptomatic SB	LA, RA, SVC	NR	18 ± 6	Significant symptom improvement	0.001	2015 [24]
HFS *	10	50.4 ± 6.4	VVS	LA	50.2 ± 3.8	36.4 ± 22.2	0	0.751	2016 [21]
AA *	47	41.7 ± 14.1	VVS	LA	43.7 ± 6.1	36.4 ± 22.2	5	0.751	2016 [21]
HFS *	43	NR	VVS	NR	NR	NR	NR	SD	2022 [56]
HFS *	40	NR	AF	NR	NR	NR	NR	SD	2022 [56]
AA *	42	51.2 ± 15.3	VVS	LA	NR	8	16%	SD	2022 [57]
HFS *	66	51.2 ± 15.3	VVS	LA	NR	8	16%	SD	2022 [57]
HFS	28	NR	VVS	LA	NR	NR	NR	SD	2021 [58]
HFS + AA	115	NR	VVS	LA	NR	21.4 ± 13.1	NR	SD	2019 [59]
AA	14	34.0 ± 13.8	VVS, advanced AVB, sinus arrest	LA, RV, LA	112 ± 15	22.5 ± 11.3	0 VVS; 0 sinus arrest; 4 AV block	0.002 (SDNN 30) days post-procedure); 0.002 (pNN > 50, 30 days post-procedure)	2017 [50]
AA	18	36.9 ± 11.2	VVS	RA	NR	34.1 ± 6.1	3 (16.6%)	0.001	2021 [60]
‡ AA	62	47.8 ± 12.6	Symptomatic SB	LA, RA, SVC	NR	12	Significant symptom improvement in	SD	2017 [61]
AA	26	41.8 ± 15.4	VVS + PVCs	LA	NR	10.6 ± 6.8	1 (3.8%)		2021 [51]
† NR	51	NR	VVS	NR	NR	22	2 (3.9%)	NR	2022 [62]
NR	123	42.2 ± 17.1	VVS	LA	NR	48 ± 13.2	33 (26.8%)	NR	2022 [63]
‡ CardNM (computed tomographic scan + electro-anatomical mapping)	20	41.4 ± 18.8	NMS	RA, SVC, coronary sinus	7 ± 4	6	4 (20%)	0.001	2018 [64]
‡ CardNM (computed tomographic scan + electro-anatomical mapping)	50	42.4 ± 17	NMS	RA, SVC, coronary sinus	8 ± 4	12	13 (26%)	0.001	2021 [65]

SA, spectral analysis; AA, anatomic approach; NMS, neurally mediated syncope; LA, left atrium’ RA, right atrium; SVC, superior vena cava; AVB atrioventricular block; FFT, fast Fourier transform analysis; NR, not reported; HFS, high-frequency stimulation; SDNN, standard deviation of all normal R-R intervals in the 24 h ECG recording; SND, sinus node dysfunction; pNN > 50, percentage of sinus cycles differing from the preceding cycle by >50 ms during the 24 h ECG recording; AF, atrial fibrillation; VVS, vasovagal syncope; PVI, pulmonary vein isolation; SB, sinus bradycardia; SD, significantly decreased; PVCs, premature ventricular contractions; CardNM, cardio-neuromodulation. * These data were taken from the same study. †: Case-control study; ‡: prospective study.

**Table 2 ijms-24-13264-t002:** Previous case studies of catheter ablation for neurally mediated syncope.

Technique	Age (y), (Sex)	Diagnosis	Approach	Procedure Time (min)	Follow-Up Duration (Months)	Syncope Recurrence	Heart Rate Variability (*p*-Value)	Year, Reference
HFS	20 (F)	VVS	RA, LA	NR	7	1	SD	2022 [66]
HFS	52 (F)	VVS	RA, LA	18.8	18	0	NR	2020 [20]
HFS (selective vagal denervation)	57 (F)	VVS	RA, LA	NR	12	0	SD	2012 [40]
HFS	15 (F)	NMS	RA, RV, LA	NR	13	3	SD	2009 [41]
Electro-anatomical mapping CARTO3	38 (M)	AVB	RA, LV, coronary sinus, His bundle, LA	NR	11	NR	NR	2021 [67]
Electro-anatomical mapping CARTO3	35 (F)	NMS and AVB	SA and AV nodes (RA, SVC, R, L, interatrial septum, LA)	NR	10	2	NR	2016 [42]
Electro-anatomical mapping CARTO3 + spectral mapping	17 (M)	NMS	SVC	NR	12	0	NR	2015 [68]
Electro-anatomical mapping CARTO3 *	31 (F)	VVS	RA, SVC, IVC, sinus AV nodes	NR	8	0	NS	2012 [69]
Electro-anatomical mapping CARTO3 *	45 (F)	VVS	RA, SVC, IVC, sinus AV nodes	NR	8	0	NS	2012 [69]
Spectral mapping (fibrillar myocardium)	23 (F)	AVB	RA, His bundle region, AV and sinus nodes	NR	21	NR	SD	2006 [70]
Spectral mapping + HFS	55 (F)	AVB	RA, SVC	75	12	0	NR	2015 [71]
Spectral mapping (fibrillar myocardium)	38 (M)	Vagal mediated AVB	Sinus nodes, AV nodes (R, L inter-atrial septum innervation)	87	15	NA	SD	2016 [39]
‡ CardNM (computed tomographic scan + electro-anatomical mapping)	16 (F)	VVS	LA	3	22	0	SD	2016 [72]
Fractionation mapping software (Ensite Precision, Abbott https://www.cardiovascular.abbott/us/en/hcp/products/electrophysiology/mapping-systems/ensite.html, accessed on 6 August 2023)	18 (M)	VVS	LA, RA	NR	NR	NR	NR	2021 [73]
Fractionation mapping software (Ensite Precision, Abbott)	20 (F)	VVS	RA	11.4	1	0	SD	2021 [74]
Fractionation mapping software (Ensite Precision, Abbott)	46 (F)	VVS	RA	NR	NR	NR	NR	2021 [75]

HFS, high-frequency stimulation; VVS, vasovagal syncope; RA, right atrium; NR, not reported; LA, left atrium; SD, significantly decreased; AVB atrioventricular block; LV, left ventricle; LA, left atrium; NMS, neurally mediated syncope; SA, sinoatrial; AV, atrioventricular; SVC, superior vena cava, IVC, inferior vena cava. * These data were taken from the same study. ‡: Prospective study.

## Data Availability

Not applicable.

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
