# Peer review of "Ganglionated Plexus Ablation Procedures to Treat Vasovagal Syncope"

_ijms, 2023, doi:10.3390/ijms241713264_

Round 1
Reviewer 1 Report
This is a nice and concise review of the field on cardioneuroablation in the context of VVS. I have very few significant remarks, but I would ask if the reviewers could perhaps include information or thoughts in relation to two things. Firstly to discuss about re-innvervation of thermally ablated ganglia and include discussion of any clinical data where VVS recurrence might be associated with re-innervation. Secondly, what are the reviewers insight/information in relation to how the introduction of pulsed field ablation (PFA) may influence development of the technique. From atrial fibrillation work it is already known that endocardial PFA does not ablate or modulate the ganglia - will this be a problem for endocardial CNA?
Additionally, Section 12 on Post ablation Validation should include discussion about extracardiac vagal stimulation (ECVS). The Pachon group has widely explored this and it is clinically viable.
I have some other very minor typographical comments:
P2 - Line 63: neurally spelled incorrectly
P5 - Line 185: inferior spelled incorrectly
Cardioneuroblation is abbreviated as CardNA in the manuscript. I think the more familiar abbreviation is CNA.
Reviewer 2 Report
This is a review manuscript by Yarkoni et al on ablation for vasovagal syncope. Overall, the manuscript is well written and offers a good review on a specific topic.
Here are some comments:
- In the Introduction, line 63, please change "neutrally" by "neurally". Also do the changes in the legends of Tables 1 and 2 (lines 324, 325, 432 and 436).
- I would use a different word than "paralysis" in in the Abstract (line 45) and Introduction (line 72). This can be misleading.
- Since pacemaker implantation can be useful in only a subgroup of patients with vasovagal syncope and significant brady-asystole, please modify the sentence in lines 74-76.
- What is the reason for "(65)" in line 203 of section 4? Is it a reference? If yes, it does not follow the other references.
- You cannot say that there are no complication with this approach (line 221). I think that it is preferable to mention "limited complications". I would also mention somewhere in the text some of the potential complications associated with GPs ablation, like cardiac perforation, sinus node dysfunction, AV block, phrenic nerve paralysis, stroke (for left-sided procedure), ...
- Please add "(2)" prior to the second subtype of vasovagal syncope on page 6, line 242.
- Please modify reference in section 7, line 274 - reference 34 versus reference 36 by Pachon.
- A visual representation of AF nests or signals (and/or technic use) would be useful for the reader in the section 7 on spectral mapping analysis.
- Please change for "Heart rate variability" in the first line of Tables 1 and 2. Also do changes to standardize Table 1 (i.e. only 1 number after decimal, yrs/min/mo only in the top of the column, ...). Why did you use a variation of grey in Table 1?
- Please change "vagolyisis" for "vagolysis" on line 388.
- I personally think that it would be useful to add a new figure with a 3D picture of the sites of ganglionated plexi (and/or ablation sites). This would help the readers for a better visualization of the different GPs.
- Duplication of "NR, not reported" in the legend of Table 2.
